# Inhibition of PCSK9 Attenuates Liver Endothelial Cell Activation Induced by Colorectal Cancer Stem Cells During Liver Metastasis

**DOI:** 10.3390/cancers17121977

**Published:** 2025-06-13

**Authors:** Ander Martin, Daniela Gerovska, Marcos J. Arauzo-Bravo, Maitane Duarte García-Escudero, Helena García García, Iratxe Bañares, Naroa Fontal, Geraldine Siegfried, Serge Evrad, Simon Pernot, Abdel-Majid Khatib, Iker Badiola

**Affiliations:** 1Bordeaux Institute of Oncology (BRIC)-UMR1312, University of Bordeaux, 33000 Bordeaux, France; ander.martinss@ehu.eus (A.M.); geraldine.siegfried@u-bordeaux.fr (G.S.); 2Department of Cell Biology and Histology, University of the Basque Country, 48940 Leioa, Spain; marcosjesus.arauzobravo@bio-gipuzkoa.eus (M.J.A.-B.); maitane.duarte@ehu.eus (M.D.G.-E.); iratxe.banares@ehu.eus (I.B.); naroa.fontal@ehu.eus (N.F.); 3Computational Biology and Systems Biomedicine, Biogipuzkoa Health Research Institute, 20014 San Sebastian, Spain; danielaivanova.gerovska@bio-gipuzkoa.eus; 4Basque Foundation for Science, IKERBASQUE, 48013 Bilbao, Spain; 5CIBER of Frailty and Healthy Aging (CIBERfes), 28029 Madrid, Spain; 6Institut Bergonié, 33076 Bordeaux, France; s.evrard@bordeaux.unicancer.fr (S.E.); s.pernot@bordeaux.unicancer.fr (S.P.)

**Keywords:** tumor microenvironment, angiogenesis, metastatic niche, protein convertase, PCSK9

## Abstract

Colorectal cancer often leads to liver metastases, a major cause of patient mortality. This study investigates the role of PCSK9, a protein lately related to the metastatic process. Researchers found that conditioned media from colorectal cancer stem cells strongly upregulates PCSK9 expression in liver sinusoidal endothelial cells (LSECs). PCSK9 activation enhances LSEC proliferation and migration, contributing to the formation of a pro-metastatic niche. Inhibiting PCSK9 with the small molecule PF-06446864 reduced endothelial activation and normalized gene expression, decreasing LSEC potential support of the metastasis. Immunofluorescence confirmed PCSK9 expression in LSECs of human colorectal cancer liver metastases. These results suggest that PCSK9 could represent a promising therapeutic target to prevent and treat liver metastases in colorectal cancer patients.

## 1. Introduction

Cancer remains a leading global cause of morbidity and mortality, with colorectal cancer (CRC) ranking among the most prevalent and lethal malignancies. In 2022, CRC accounted for nearly 2 million new cases worldwide and a substantial proportion of cancer-related deaths [1]. While CRC originates in the colon or rectum, most CRC-related mortality stems from its metastatic potential [2]. Among metastatic sites, the liver is the most frequently affected organ due to its special anatomical, physiological, and immunological characteristics [3].

The liver’s dual blood supply, via the portal vein and hepatic artery, facilitates the entrapment of circulating tumor cells from the gastrointestinal tract [4]. The sinusoidal vasculature, lined by liver sinusoidal endothelial cells (LSECs), is particularly conducive to tumor cell adhesion and extravasation. LSECs are pivotal in the metastatic cascade, promoting tumor cell attachment, penetration into the hepatic parenchyma, and immune evasion. Tumor cells exploit LSECs through adhesion molecules such as intercellular adhesion molecule-1 (ICAM-1) and vascular cell adhesion protein-1 (VCAM-1), aiding liver colonization [5,6]. Additionally, LSECs contribute to the liver’s immunosuppressive microenvironment by expressing programmed death-ligand 1 (PD-L1), which inhibits cytotoxic T-cell activation and enables immune evasion [7,8,9].

Beyond immune modulation, LSECs actively shape the hepatic metastatic niche [10]. These specialized endothelial cells overexpress proteins such as LSEC-derived extra domain A or fibronectin, promoting epithelial-to-mesenchymal transition [11]. LSECs also regulate microRNAs involved in cellular behavior and tumor progression [12]. Their unique structural features, including fenestrations that provide access to the basement membrane, enhance tumor cell invasion and the establishment of metastatic lesions [13,14]. Furthermore, the liver’s regenerative capacity and immunosuppressive environment increase its susceptibility to metastases, underscoring its importance as a therapeutic target in metastatic CRC [15].

Emerging evidence has identified proprotein convertase subtilisin/kexin type 9 (PCSK9) as a key player in CRC progression and liver metastasis [16,17]. Initially characterized for its role in cholesterol metabolism, PCSK9 is now recognized as a regulator of critical processes in cancer metastasis, including epithelial–mesenchymal transition (EMT), immune evasion, and endothelial activation [18,19]. PCSK9 reduces E-cadherin expression, increases N-cadherin levels, and enhances matrix metalloproteinase-9 (MMP9) activity, thereby promoting tumor cell migration and invasion [16]. It also impairs immune recognition by downregulating major histocompatibility complex class I (MHC-I) expression on tumor cells, facilitating immune evasion [20,21].

In the liver, PCSK9 plays a crucial role in LSEC. It has been shown that tumor-derived signals upregulate PCSK9 expression, supporting vascular co-option—a mechanism allowing tumor cells to reorganize independently of angiogenesis [22,23]. This resistance to anti-angiogenic therapies highlights the potential of PCSK9 as a therapeutic target.

Given PCSK9’s involvement in the processes described above, it is of particular interest to investigate its role in liver sinusoidal endothelial cells (LSECs). Understanding this interaction could contribute to the development of therapies aimed at mitigating colorectal cancer liver metastasis and potentially enhancing the efficacy of current anti-angiogenic treatments.

## 2. Materials and Methods

### 2.1. Cell Culture and Cancer Stem Cell Derivation

Immortalized liver sinusoidal endothelial cells (LSECs) from both human (hLSEC) and murine (mLSEC) origins were obtained from Innoprot (Innoprot, Derio, Spain). These cells were cultured as adherent monolayers on culture plates pretreated with rat-tail type I collagen (08-115, Sigma Aldrich, St. Louis, MI, USA) for 1 h at room temperature, at 37 °C and 5% CO_2_. Complete Endothelial Cell Medium (P60104, Innoprot) was used to maintain the cultures. The human metastatic colorectal adenocarcinoma cell line SW620 and the murine metastatic colorectal adenocarcinoma cell line CT26 were acquired from the American Type Culture Collection (ATCC). SW620 cells were maintained in Leibovitz’s L15 medium (L5520, Sigma Aldrich), while CT26 cells were cultured in fully supplemented RPMI medium (R8758, Sigma Aldrich). To derive cancer stem cell (CSC)-like populations from SW620 and CT26 cells, 1 × 10^4^ cells per well were seeded in low-adherence 6-well plates and cultured for 7 days in serum-free DMEM-F12 medium (11554546, GIBCO). The medium was supplemented with 1% *v*/*v* N2, 0.5% *v*/*v* B27, 20 ng/mL fibroblast growth factor (FGF), and 20 ng/mL epidermal growth factor (EGF). Conditioned media were obtained after incubation of cell cultures for 24 h with non-complete Endothelial Cell Media. Conditioned media was diluted (1:1 *v*/*v*) with complete media before being added to the cells. To obtain the condition activated with CSC-conditioned media in the absence of PCSK9, LSECs were pretreated 24 h before stimulation with 50 µg/mL PCSK9 chemical inhibitor (PF-06446864, Sigma-Aldrich, St. Louis, MO, USA).

### 2.2. RNA Extraction and Quantitative Real-Time PCR

To compare PCSK9 expression in LSEC and CSC markers in colon cancer cells, total RNA isolation from cell cultures was performed using the NucleoSpin RNA isolation kit (740955, Macherey & Nagel, Düren, Germany). The RNA was reverse transcribed using the iScript cDNA synthesis kit (1708891, BioRad, Hercules, CA, USA) according to the manufacturer’s guidelines and was used for real-time PCR. Real-time PCR was performed using the Power SYBR^®^ Green Master Mix (1725271, BioRad) and the primers shown in Table 1. The quantitative polymerase chain reaction (qPCR) data were acquired with the CFX96 C1000 Touch Real-Time PCR Detection System (Biorad). The expression levels were normalized to β18S ribosomal RNA (18S rRNA). All the reactions were performed in triplicate, and the relative expression of each gene was calculated via the 2^−∆∆Ct^ method [24].

### 2.3. Total RNA Sequencing

After performing quality control, checking that the RIN value of each sample was bigger than 6, and ensuring RNA quantity was higher than 200 ng, library construction was performed following Illumina’s recommendation. The sequencing library was prepared by random fragmentation of the cDNA sample, followed by 5′ and 3′ adapter ligation. The fragmentation and ligation reactions were performed in a single step that increases the efficiency of the library preparation process. Then, adapter-ligated fragments were amplified by PCR and purified in a gel. Subsequently, the library was loaded into a flow cell where fragments were captured on a lawn of surface-bound oligos complementary to the library adapters. Each fragment was then amplified into distinct clonal clusters through bridge amplification. When cluster generation was completed, sequencing was performed using Illumina SBS technology. All sequencing data was converted into raw data in FASTQ format, allowing its bioinformatics analysis.

### 2.4. Protein Extraction and Western Blot

LSECs were washed twice with cold 1× PBS and subsequently lysed using RIPA buffer at a concentration of 100 µL per 10^6^ cells. Protein concentrations were determined using the Bicinchoninic Acid (BCA) assay (Sigma-Aldrich). Samples were resuspended in 4x loading buffer (250 mM Tris-HCl pH 6.8, 500 mM β-mercaptoethanol, 50% glycerol, 10% SDS and bromophenol blue). For each sample, 20 µg of protein lysate was loaded onto a 10% SDS-PAGE gel and electrophoresed for 1 h and 30 min at 90 V. Proteins were then transferred onto a nitrocellulose membrane under wet transfer conditions at a constant current of 385 mA for 3 h and 30 min, with cooling on ice. Following protein transfer, membranes were incubated overnight at 4 °C with primary antibodies, including rabbit anti-PCSK9 (1:2000, ab181142, Abcam) and mouse anti-GAPDH (1:1000, MCA4739, Biorad). Subsequently, the membranes were incubated with horseradish peroxidase (HRP)-conjugated secondary antibodies (anti-mouse HRP (ab6728, Abcam, Cambridge, UK) and anti-rabbit HRP (ab102279, Abcam) for 1 h and 30 min at room temperature. Protein detection was performed using Immobilon Crescendo Western HRP substrate (WBLUR0100, Merck, Darmstadt, Germany), and chemiluminescence signals were visualized and captured using the iBright CL1500 Imaging System (Invitrogen, Thermo Fisher Scientific, Waltham, MA, USA). GAPDH protein levels were used as a loading control to normalize protein expression data.

### 2.5. Immunofluorescent Staining

LSECs were seeded on collagen-coated coverslips, fixed with 4% paraformaldehyde (PFA), and permeabilized with PBS-Triton X-100 (X100, SigmaAldrich) and BSA (A9418, Sigma Aldrich). After blocking, cells were incubated overnight with primary antibodies, followed by fluorophore-conjugated secondary antibodies and DAPI for nuclear staining. Coverslips were mounted with Fluoromount-G, and imaging was performed using a Zeiss LSM800 confocal microscope (Zeiss, Oberkochen, Germany)with ImageJ software (v1.54p) for processing. Regarding tissue immunofluorescent staining, liver tissue sections were prepared using a cryostat and fixed with 4% PFA. After permeabilization and blocking, tissues were incubated with primary antibodies overnight, followed by secondary antibodies and DAPI. To reduce autofluorescence, tissues were treated with Sudan Black B and ethanol. Finally, slides were mounted with Fluoromount-G and stored at 4 °C for long-term use. Used antibodies were anti-CD31 mouse (Dako, M0823), anti-CD146 rabbit (Abcam, ab75769) and anti-PCSK9 goat (Abcam, ab28770). Image processing was performed using Image J software ((Version 1.54p). The representations appearing in the results were obtained using the Z-stack option of this software.

### 2.6. Cell Proliferation Assay

The effect of PCSK9 inhibition on liver sinusoidal endothelial cell (LSEC) proliferation was assessed using Crystal Violet staining (C0775, Sigma Aldrich). mLSECs and hLSECs were seeded in 96-well plates and treated for 24 h with a 50 µg/mL PCSK9 chemical inhibitor. Subsequently, conditioned media were added to activate the cells for another 24 h. Cells were stained with 0.5% Crystal Violet for 20 min, washed, and dried for 24 h. Methanol was added to extract the stain, and plates were incubated for 20 min with gentle agitation. The optical density at 570 nm (OD570) was measured using a microplate reader, with background OD from empty wells subtracted. Non-treated cells served as the control (100% viability), and the percentage of viable treated cells was calculated based on OD570 values.

### 2.7. Cell Migration Assay

The effect of PCSK9 inhibition on LSEC migration was assessed using a wound-healing assay. hLSECs and mLSECs were seeded in 48-well plates, treated with a 50 µg/mL PCSK9 inhibitor for 24 h, and then incubated with conditioned media for another 24 h. To ensure only migration was analyzed, cell proliferation was blocked with Mitomycin C (M4287, Merck) at 1 µg/mL for 2 h. A scratch was created using a pipette tip, and wells were washed with PBS to remove debris before being refilled with fresh media. Images of the scratch were captured at 0 and 4 h using a Zeiss Primovert microscope. Migration rates were determined by calculating the percentage of scratch closure between the two time points.

### 2.8. Statistical Analysis

Statistical analyses were performed using the GraphPad Prism software (Version 9.2.0, San Diego, CA, USA). Unless otherwise indicated, a parametric Student *t*-test was applied to compare differences between two groups. For experiments with more than 2 groups and only one variable, One-way ANOVA and Tukey’s post hoc test were performed. If the results studied did not show a normal distribution, the Kruskal–Wallis test was performed. If there were more than 2 groups and 2 variables to compare, a two-way ANOVA and a Tukey post hoc test were performed. Differences of *p* < 0.05 were considered statistically significant (*, *p* < 0.05; **, *p* < 0.01; ***, *p* < 0.001). Unless otherwise stated, data are expressed as mean ± standard error deviation (SD).

## 3. Results

### 3.1. Endothelial and Cancer Cell Culture Validation

To elucidate the role of LSECs in colorectal liver metastasis, the endothelial phenotype of LSECs was verified through immunofluorescent staining for endothelial markers CD31 and CD146, as shown in Figure 1A. Both markers were prominently expressed, confirming the endothelial identity of the analyzed cell population.


Figure 1Endothelial cell and cancer stem cell phenotype validation. (**A**) Both mouse and human LSECs present CD31 and CD146 expression by immunofluorescent staining; (**B**) SW620 and CT26 cancer cells show an adherent phenotype, while both cell types after reprogramming show a non-adherent phenotype, forming spheres; (**C**–**E**) reprogrammed CSCs show an increased expression at the mRNA level of pluripotency genes SOX2, EpCAM and PROM1 compared to the cancer cells CT26 and SW620. Unpaired *t*-test (*n* = 4).
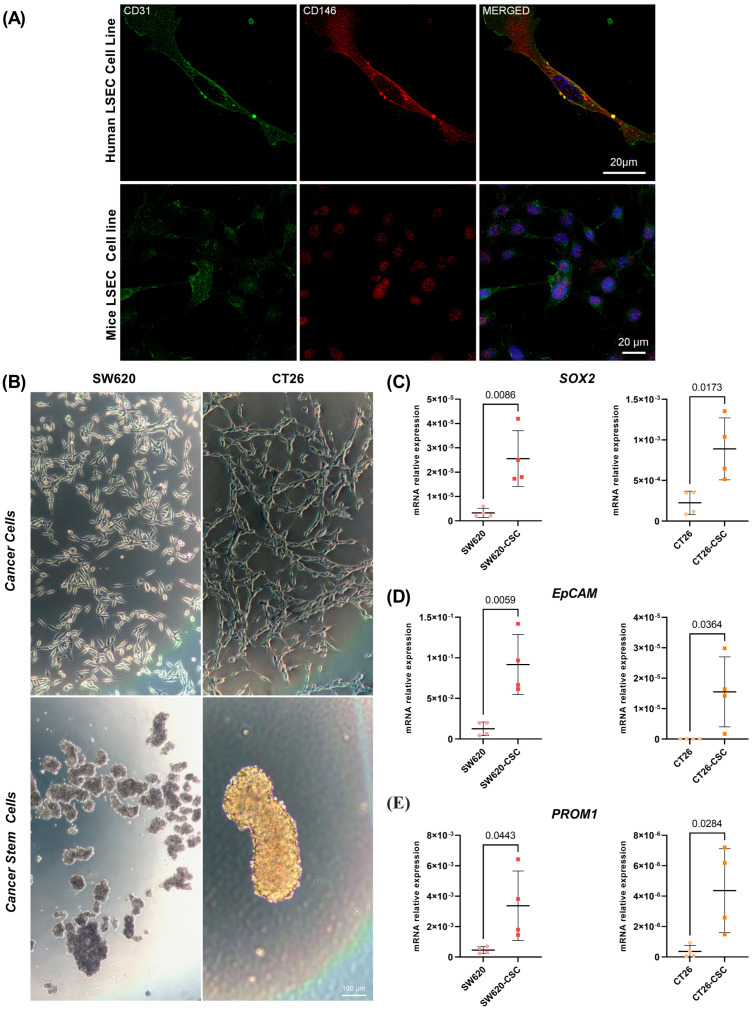



Similarly, to investigate differences in endothelial activation induced by conditioned media from CSCs versus differentiated cancer cells, the CSC phenotype was first validated. As shown in Figure 1B, CSCs exhibited the ability to form floating spheres, whereas parental cells displayed an adherent growth pattern. Furthermore, the expression levels of stem cell markers, including Sox2, EpCAM, and CD133, were assessed (Figure 1C–E). The analysis revealed significantly higher expression of these genes in CSCs compared to differentiated cancer cells, thereby confirming their stem cell-like phenotype.

### 3.2. Human LSEC Total RNA Sequencing Reveals PCSK9 Overexpression After Activation with CSC Conditioned Media

To investigate gene expression variation in hLSECs, cells were activated with conditioned media from SW620 parental cells and SW620-CSC tumor stem cells. Figure 2A,B demonstrate that stimulation of LSECs with conditioned media derived from SW620 cancer stem cells (SW620-CSC) leads to a marked overexpression of genes predominantly located on chromosomes 1 and 12. Notably, as shown in Figure 2C,D, among the genes located on chromosome 1, PCSK9 is of particular interest as it is significantly upregulated and ranks among the top 50 most overexpressed genes in LSECs following stimulation with SW620-CSC conditioned media. This highlights PCSK9’s potential involvement in LSEC activation by cancer stem cells.

### 3.3. Human and Mice LSEC Cultures Show PCSK9 Overexpression After CSC Conditioned Media Activation

To validate the total RNA sequencing findings, LSECs were activated with conditioned media. Notably, CSC-conditioned media significantly upregulated PCSK9 mRNA levels in both human (Figure 3A) and murine LSECs (Figure 3C). Western blot analysis confirmed PCSK9 expression at the protein level under studied conditions. Even more, the relative ban intensity measure revealed an overexpression of PCSK9 after activation of murine and human LSEC with CSC-conditioned media (Figure 3B–D). Additionally, Figure 3E shows an immunofluorescent staining of CD31 and PCSK9 in colorectal cancer liver metastasis patient tissues revealed co-staining, indicating PCSK9 expression in patient LSECs.

### 3.4. PCSK9 Inhibition Downregulates microRNA Mediated Gene Silencing in Activated Human LSEC

To investigate gene expression changes in human liver sinusoidal endothelial cells in the presence or absence of PCSK9 activity, RNA sequencing was conducted on hLSECs pre-treated with a PCSK9 inhibitor for 24 h prior to stimulation with SW620-CSC conditioned media. This experimental design enabled a direct comparison between PCSK9-inhibited and untreated cells under identical stimulation conditions.

Figure 4A,B highlight the most significantly downregulated genes following PCSK9 inhibition. Among the most prominent transcriptional changes were microRNAs. As shown in Figure 4C, a specific group of microRNAs associated with post-transcriptional gene silencing emerged as particularly relevant. Furthermore, the volcano plot in Figure 4D reveals that three microRNAs—MIR93, MIR30C, and MIR194-2HG—were significantly upregulated in LSECs stimulated with SW620-CSC conditioned media, while their expression was notably reduced following PCSK9 inhibition. These findings suggest a potential regulatory role for PCSK9 in microRNA-mediated gene expression pathways in LSECs.

### 3.5. LSEC Proliferation Triggered by CSC Media Is Reduced After PCSK9 Inhibition with PF-06446864

To investigate the role of PCSK9 in LSECs during the angiogenic process and to link it with microRNA downregulation, we began by assessing LSEC proliferation rates. As shown in Figure 5A–C, activation with CSC-conditioned media led to a significant increase in the proliferation rates of both human and murine LSECs, whereas activation with parental-conditioned media did not induce a similar effect.

Building upon this observation, we treated LSECs with a PCSK9 inhibitor and measured their proliferation rates. In human LSECs, PCSK9 inhibition did not significantly alter proliferation under control conditions. However, in cells activated with SW620-conditioned media, the proliferation rate decreased significantly, with an even greater reduction observed in cells activated with CSC-conditioned media (Figure 5B).

In contrast, murine LSECs exhibited a more pronounced response to PCSK9 inhibition across all conditions. Specifically, PCSK9 inhibition resulted in a significant reduction in proliferation rate, including under control conditions, as well as in cells activated with both parental- and CSC-conditioned media (Figure 5D). This suggests a stronger effect of PCSK9 inhibitor in murine LSECs compared to human ones.

### 3.6. LSEC Migration Triggered by CSC Media Is Reduced After PCSK9 Inhibition with PF-06446864

Looking deeper at PCSK9’s role in the angiogenic process, we next examined its effect on the migration of human and murine LSECs. Similarly to proliferation, both human and murine LSECs exhibited a significant increase in migration rates following activation with CSC-conditioned media (Figure 6A–C). Notably, murine LSECs demonstrated faster migration rates compared to human ones.

When examining the impact of PCSK9 inhibition in Figure 6B, we observed distinct effects in human and murine LSECs. In human cells, PCSK9 inhibition significantly reduced migration rate in cells activated with both SW620 and CSC-conditioned media, with the reduction being more pronounced in CSC-activated cells.

In Figure 6D, murine LSECs, PCSK9 inhibition led to a significant decrease in migration rates under all three conditions, with the strongest reduction observed in cells activated with CT26 and CSC-conditioned media. Consistent with our findings on proliferation, murine LSECs appeared more sensitive to PCSK9 inhibition than human cells also in migration.

## 4. Discussion

PCSK9, first described in 2003 by Seidah et al., has been extensively studied for its role in hypercholesterolemia, as it promotes LDL receptor (LDLR) degradation through intracellular and extracellular pathways. Initially linked only to cholesterol metabolism, PCSK9 has recently been found to be overexpressed in various tumors, including colorectal carcinoma. Emerging research suggests PCSK9 also degrades MHC-I, reducing tumor immunogenicity and restricting lymphocyte activity [20,21,25]. Furthermore, recent studies have shown that PCSK9 is directly implicated in the metastatic progression of both breast and pancreatic cancers [26,27]. While most studies focus on PCSK9 in tumor cells, its role in the tumor microenvironment remains largely unexplored. This study aims to investigate PCSK9’s function in LSECs within colorectal liver metastases.

PCSK9 remains largely unexplored in liver sinusoidal endothelial cells (LSECs) under both physiological and pathological conditions. Databases like the Human Protein Atlas report basal PCSK9 expression in LSECs [28], but our study confirms its presence at both mRNA and protein levels in human and mice LSEC lines.

We found that activation with metastatic colorectal carcinoma stem cell (CSC)-conditioned media significantly upregulates PCSK9 in LSECs, while media from parental tumor cells does not. This difference may be linked to the metastatic process, as CSCs play a key role in tumor growth and angiogenesis. CSCs initially lose their LGR5+ phenotype to acquire a migratory EMP1+ state, allowing metastasis. Once in the liver, these cells revert to LGR5+, initiating tumor expansion and microenvironment remodeling, including angiogenesis [29]. Since angiogenesis begins only when metastases exceed 1 mm^3^, it is likely that CSCs drive LSEC activation to support metastatic growth [30].

Although PCSK9 has been extensively studied for its role in LDLR [18,31] and immune modulation, little is known about its function in LSECs. Thus, after establishing the overexpression of PCSK9, we investigated the pathway through which this protein might be influencing LSECs. To this end, we performed RNA sequencing (RNAseq) on hLSECs, analyzing two conditions: hLSECs activated with SW620-CSC media and the same cells pretreated with a PCSK9 chemical inhibitor, PF-06446864. This inhibitor was found to inhibit PCSK9 with a high specificity, avoiding overtargets [32]. The goal was to identify which pathways were altered upon PCSK9 inhibition in activated hLSECs. Our analysis revealed that, among several gene clusters, microRNAs such as MIR93, MIR30C2 and MIR194-2HG, which are implicated in gene silencing, were downregulated after PCSK9 inhibition.

MIR-93 has been shown to interact with and inhibit the p21 gene, a key regulator of the cell cycle [33]. p21 suppresses CDK2, CDK1, and CDK4/6, leading to cell cycle arrest in the G1 and S phases [34,35,36]. This can suggest a possible mechanism for the increased proliferation observed when endothelial cells are activated with CSC media. Thus, the overexpression of MIR-93 may inhibit p21 production, disrupting cell cycle regulation and promoting proliferation. However, PCSK9 inhibition could reduce MIR-93 levels, restoring p21 expression and re-establishing normal cell cycle control.

MIR194-2HG has been shown to be upregulated in cancers such as cervical adenocarcinoma and bladder cancer [37,38]. Furthermore, Xu et al. demonstrated that elevated levels of this microRNA are associated with poor prognosis in liver cancers, and its overexpression in HepG2 and Huh7 cells promotes proliferation, migration, and invasion through activation of the Wnt/β-catenin signaling pathway [39]. This may explain the increased migration and proliferation observed in LSECs upon activation with CSC media. Moreover, the reduction in MIR194-2HG following PCSK9 inhibition could account for the decreased proliferation and migration observed.

Finally, although MIR30C has been described as an antitumoral microRNA and as a regulator that reduces liver sinusoidal endothelial cells during liver fibrosis, little is known about the specific MIR30C2 [40,41]. Further research is needed to determine its involvement in liver sinusoidal endothelial cells during colorectal liver metastasis.

Although the present study provides a foundation for investigating the role of PCSK9 in liver sinusoidal endothelial cells (LSECs), it has certain limitations, most notably, the inability to use the PCSK9 inhibitor PF-06446864 in vivo due to the lack of a delivery system capable of specifically targeting LSECs in this model.

In summary, this study describes that activation with CSC-conditioned media significantly increases PCSK9 expression in liver sinusoidal endothelial cells at the mRNA level. Even more, the presence of PCSK9 in LSEC is shown also at the protein level by Western blotting in cell cultures and by immunofluorescent staining in colorectal cancer liver metastasis patient tissues.

RNA sequencing revealed that PCSK9 inhibition leads to the downregulation of three microRNAs, among which MIR93 and MIR194-2HG appear to play key roles in regulating proliferation and migration during tumor progression. To confirm this, LSECs were activated with CSC-conditioned media, resulting in a significant increase in migration and proliferation. However, this effect was reversed upon PCSK9 inhibition in both human and murine LSECs.

These findings indicate that PCSK9 may be involved in promoting proliferation and migration during liver metastasis. Even more, RNA sequencing analysis showed us potential pathways altered as microRNAs such as MIR93 and MIR194-2HG. Therefore, even if more studies are needed to fully understand LSEC’s PCSK9 mechanism, its inhibition appears as a potential new therapeutic target for the treatment of colorectal liver metastasis, which could synergize with existing anti-angiogenic therapies.

## Figures and Tables

**Figure 2 cancers-17-01977-f002:**
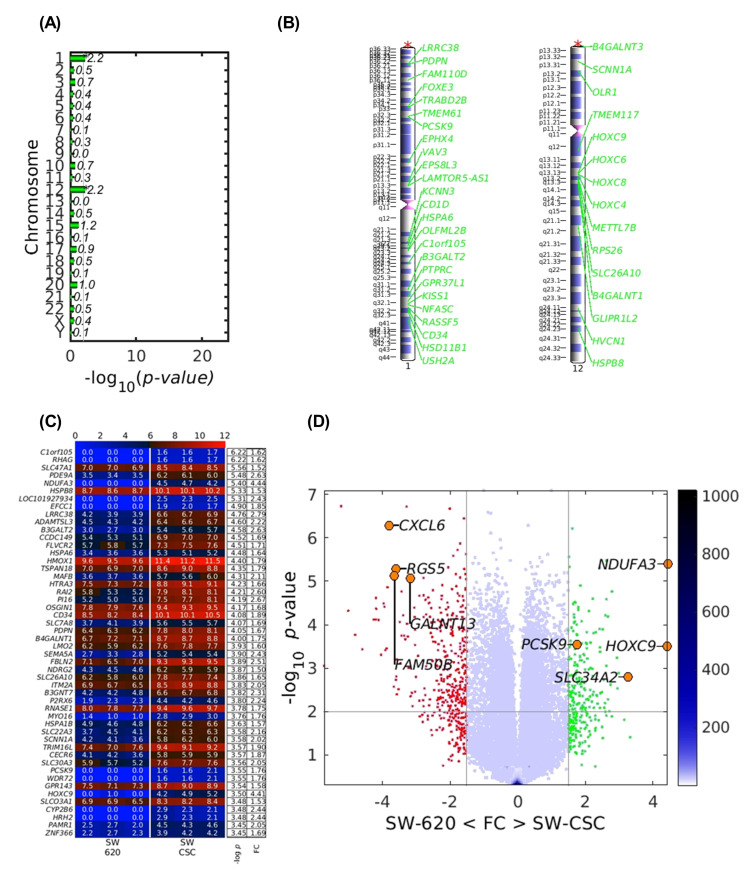
Transcriptomic profiling of LSECs stimulated with conditioned media from SW620 and SW620 cancer stem cells (SW620-CSC). (**A**) LSECs stimulated with SW620-CSC conditioned media exhibit a notable enrichment of differentially expressed genes located on chromosomes 1 and 12 (* *p* < 0.05). (**B**) Gene enrichment analysis highlights distinct transcriptional changes in LSECs exposed to SW620-CSC conditioned media, with pronounced enrichment observed on chromosomes 1 and 12. (**C**) Heatmap illustrating the top 50 upregulated genes in LSECs treated with SW620-CSC conditioned media relative to those treated with SW620 conditioned media. (**D**) Volcano plot depicting the distribution of differentially expressed genes in LSECs stimulated with SW620 versus SW620-CSC conditioned media.

**Figure 3 cancers-17-01977-f003:**
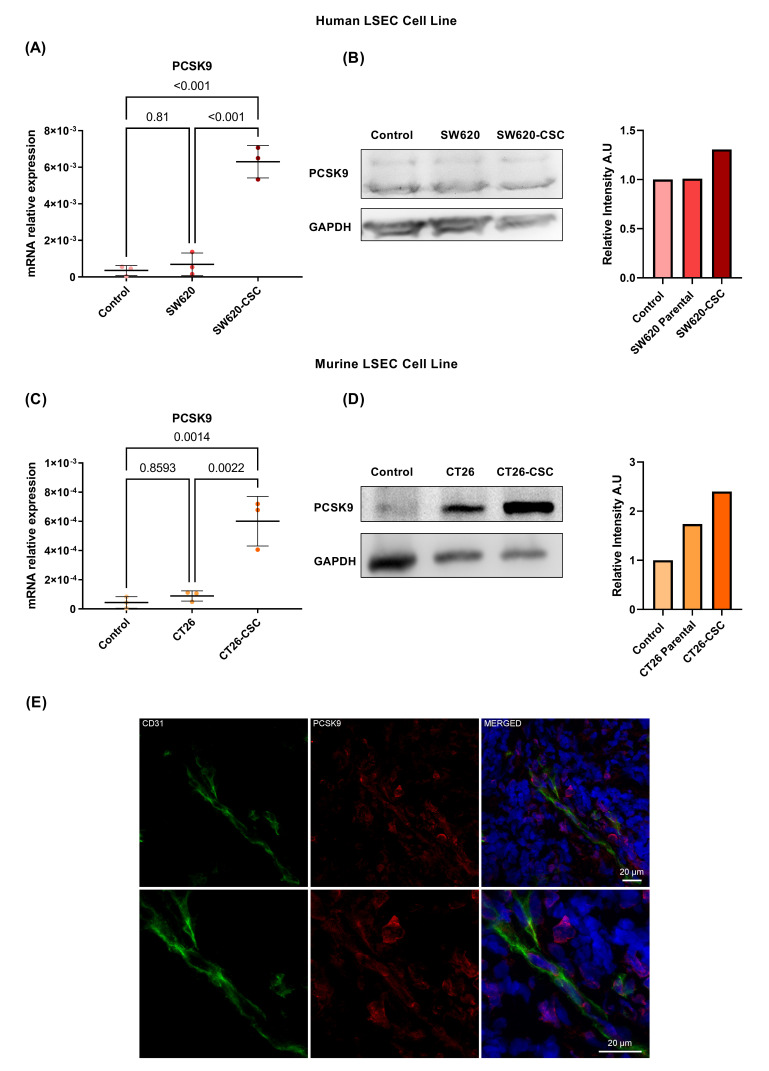
(**A**) The hLSECs show an increase in PCSK9 expression at the mRNA level when activated with SW620-CSC conditioned media compared to basal. One-way ANOVA (*n* = 3), Tukey’s post hoc test (**B**) regarding hLSEC’s PCSK9, protein expression was observed in all 3 study conditions. PCSK9 relative band intensity was measured. (uncropped Wb available in Appendix A); (**C**) mLSECs show an increase in PCSK9 expression at the mRNA level when activated with SW620-CSC conditioned media compared to basal. One-way ANOVA (*n* = 3), Tukey’s post hoc test (**D**) regarding mLSEC’s PCSK9, protein expression was observed in all 3 study conditions. PCSK9 relative band intensity was measured. (uncropped Wb available in Appendix A); (**E**) immunofluorescent staining of CD31 (green) and PCSK9 (red) in patient liver metastasis tissue shows colocalization of both markers.

**Figure 4 cancers-17-01977-f004:**
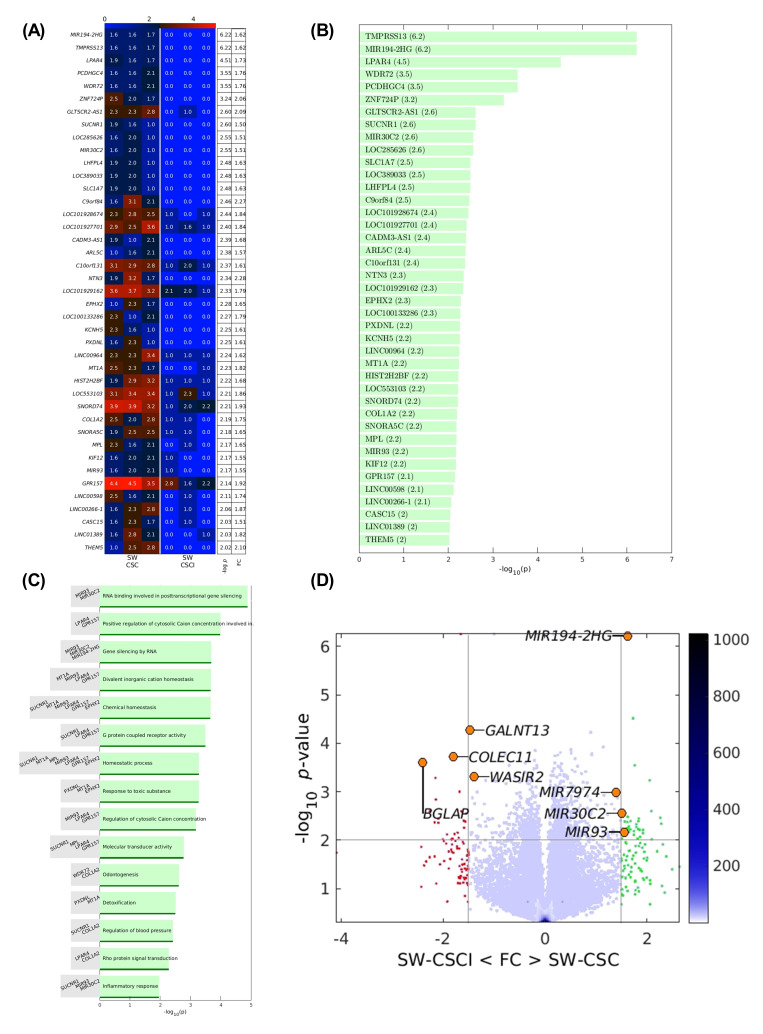
Transcriptomic profiling of LSECs treated with PCSK9 inhibitor and stimulated with conditioned media from SW620 cancer stem cells. (**A**) Heatmap illustrating the top downregulated genes in LSECs treated with PCSK9 inhibitor and stimulated with SW620-CSC conditioned. (**B**) Bar plot representing the top downregulated genes in LSECs treated with PCSK9 inhibitor and stimulated with SW620-CSC conditioned. (**C**) Bar plot categorizing the downregulated genes into functional gene families following PCSK9 inhibition. (**D**) Volcano plot showing the distribution of differentially expressed genes in LSECs stimulated with SW620-CSC conditioned media alone versus LSECs pre-treated with the PCSK9 inhibitor and subsequently stimulated with SW620-CSC conditioned media.

**Figure 5 cancers-17-01977-f005:**
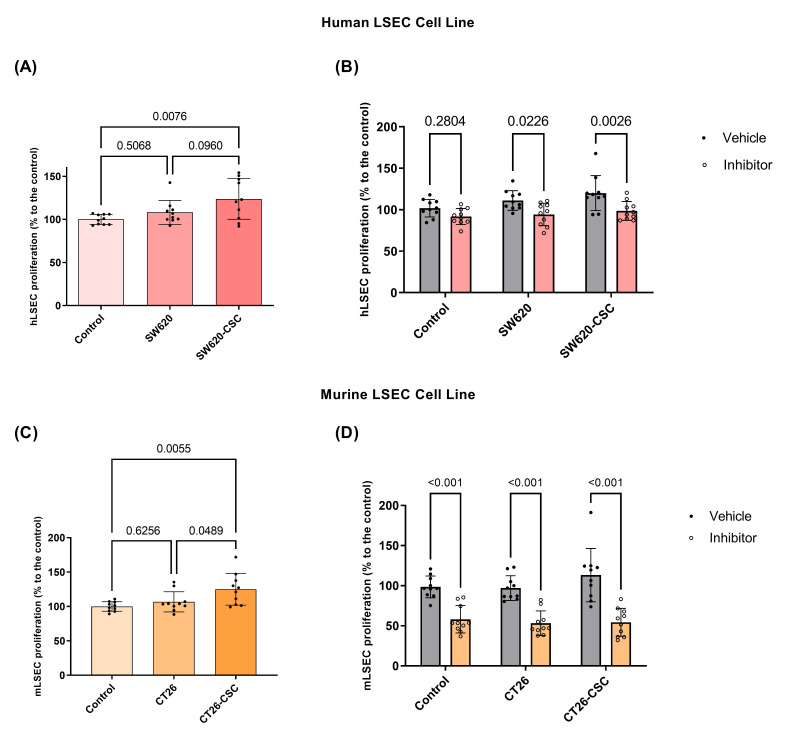
Study of the effect of conditioned media of differentiated tumor cells and tumor stem cells on hLSECs and mLSECs proliferation and of the involvement of PCSK9 in this process. (**A**) Study of the effect of SW620 and SW620-CSC conditioned media on the proliferation of hLSECs normalized to their basal condition. Kruskal–Wallis test (*n* = 10), Dunn’s post hoc test (**B**) effect of PCSK9 inhibitor and its vehicle (DMSO) on the proliferation of hLSECs in their basal condition and activated with SW620 and SW620-CSC media. Two-way ANOVA (*n* = 10), Tukey’s post hoc test (**C**) study of the effect of CT26 and CT26-CSC conditioned media on the proliferation of mLSECs normalized to their basal condition. Kruskal–Wallis test (*n* = 10), Dunn’s post hoc test (**D**) effect of PCSK9 inhibitor and its vehicle (DMSO) on the proliferation of mLSECs in their basal condition and activated with CT26 and CT26-CSC media. Two-way ANOVA (*n* = 10), Tukey’s post hoc test.

**Figure 6 cancers-17-01977-f006:**
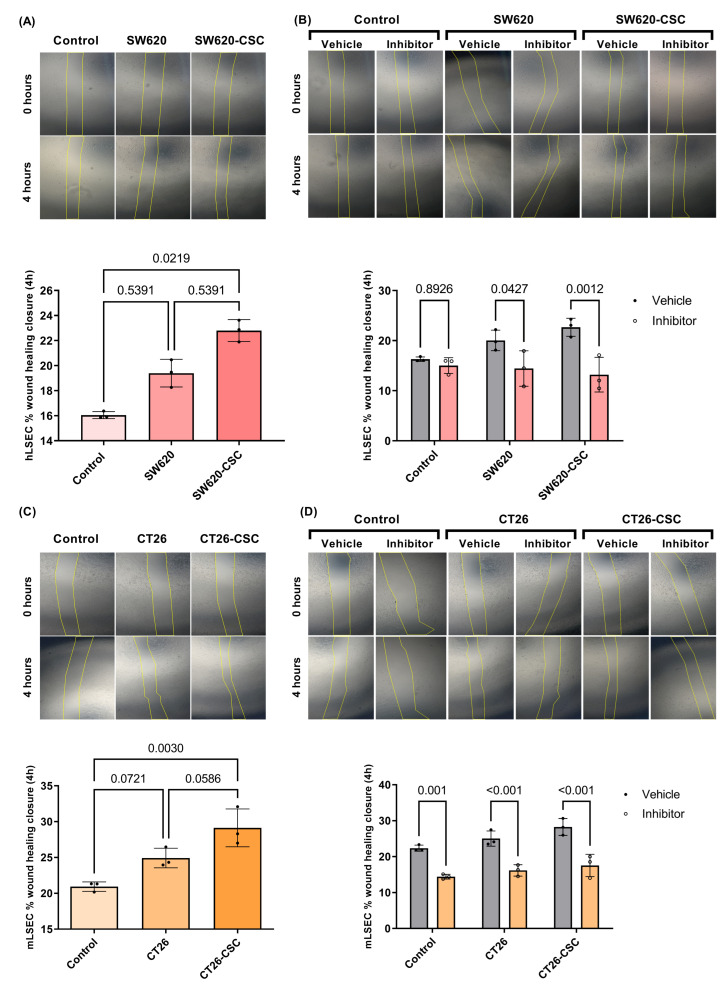
Study of the effect of conditioned media of differentiated tumor cells and tumor stem cells on hLSECs and mLSECs migration and of the involvement of PCSK9 in this process. (**A**) Study of the effect of SW620 and SW620-CSC conditioned media on the migration of hLSECs. Kruskal–Wallis test (*n* = 3), Dunn’s post hoc test (**B**) Effect of PCSK9 inhibitor and its vehicle (DMSO) on the migration of hLSECs in their basal condition and activated with SW620 and SW620-CSC media. Two-way ANOVA (*n* = 3), Tukey’s post hoc test (**C**) Study of the effect of CT26 and CT26-CSC conditioned media on the migration of mLSECs. Kruskal–Wallis test (*n* = 3), Dunn’s post hoc test. (**D**) Effect of PCSK9 inhibitor and its vehicle on the migration of mLSECs in their basal condition and activated with CT26 and CT26-CSC media. Two-way ANOVA (*n* = 3), Tukey’s post hoc test.

**Table 1 cancers-17-01977-t001:** Human and mice primers.

**Gene (Human)**	**Forward Primer Sequence (5′-3′)**	**Reverse Primer Sequence (5′-3′)**
*PCSK9*	5′-GGAGTGGTCTAGAGCCCGA-3′	5′-GCTCACACACTCGCTTGAAC-3′
*EPCAM*	5′-GAACACTGCTGGGGTCAGAA-3′	5′CTGAAGTGCAGTCCGCAAC-3′
*SOX 2*	5′-ATAATAACAATCATCGGCGG-3′	5′-AAAAAGAGAGAGGCAAACT-3′
*PROM1*	5′-AAGCATTGGCATCTTCTATG-3′	5′-TTTGCTCTGGAGTTTCATTC-3′
*RPS18*	5′CAGAAGGATGTAAAGGATGG-3′	5′TATTTCTTCTTGCACACACC-3′
Gene (Mouse)	Forward primer sequence (5′-3′)	Reverse primer sequence (5′-3′)
*PCSK9*	5′-TTGCAGCAGCTGGGAACTT-3′	5′-CCGACTGATGACCTCTGGA-3′
*EPCAM*	5′-AACACAAGACGACGTGGACA-3′	5′-GCTCTCCGTTCACTCTCAGG-3′
*SOX 2*	5′-CACAACTCGGAGATCAGCAA-3′	5′-CCTCGGGAAGCGTGTACTTA-3′
*PROM1*	5′-AGAGACCCTAGAGGACGCAA-3′	5′-GAGTGATTGGGTGAGAGCCC-3′
*RPS18*	5′-CCCTGAGAAGTTCCAGCACA-3′	5′-TGTACTGTCGTGGGTTCTGC-3′

## Data Availability

The raw and processed sequencing data generated in this study have been deposited in the NCBI Gene Expression Omnibus (GEO) under accession number GSE299058.

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
