# Peer review of "Inhibition of PCSK9 Attenuates Liver Endothelial Cell Activation Induced by Colorectal Cancer Stem Cells During Liver Metastasis"

_cancers, 2025, doi:10.3390/cancers17121977_

Round 1

Reviewer 1 Report

Comments and Suggestions for Authors

This study nicely addresses the role of PCSK9 in LSECs in context of colorectal cancer liver metastasis indicating a novel approach. But there are few questions that need to be addressed to strengthen the manuscript.

1.  The discussion part often suggests a direct causal relationship (e.g., “PCSK9 is directly implicated in encouraging proliferation and migration”), which is not entirely supported by the available evidence. 

2.  The discussion lacks a critical assessment of whether the observed effects are due to the  factors secreted by CSC or from general stress responses in LSECs. Author should discuss this in limitations or should give explanation.

3.  The findings suggest a more significant decrease in proliferation in murine LSECs as opposed to human LSECs following the inhibition of PCSK9. How do the results observed in mouse LSECs relate to human colorectal cancer liver metastasis, and how patients get benefit from this study? Moreover, in order to enhance translational relevance, have the authors analyzed mRNA or microRNA expression profiles in tissue samples derived from patients? Additionally, have any studies been performed using primary human liver LSECs isolated from either normal or tumor-adjacent liver tissue to confirm the results seen in the cell lines?

Author Response

Comments 1: 1.  The discussion part often suggests a direct causal relationship (e.g., “PCSK9 is directly implicated in encouraging proliferation and migration”), which is not entirely supported by the available evidence.

Response 1: Thank you for highlighting this issue. We agree that the current wording may overstate the strength of the evidence. We have revised the discussion to remove any definitive statements suggesting a direct causal relationship between PCSK9 and LSEC proliferation and migration. These changes have been implemented on page 17.

Comment 2: The discussion lacks a critical assessment of whether the observed effects are due to the factors secreted by CSC or from general stress responses in LSECs. Author should discuss this in limitations or should give an explanation.

Response 2: Thank you for this comment, we certainly appreciate it. To address this concern, we compared the effects of conditioned media derived from both differentiated colorectal cancer cells and cancer stem cells (CSCs). Notably, the changes in LSEC behavior—particularly PCSK9 overexpression—were more pronounced in response to CSC-derived media compared to media from differentiated cells or untreated controls. This suggests that the observed effects are specific to factors secreted by CSCs, rather than a generalized stress response due to the conditioned media itself.

Comment 3: The findings suggest a more significant decrease in proliferation in murine LSECs as opposed to human LSECs following the inhibition of PCSK9. How do the results observed in mouse LSECs relate to human colorectal cancer liver metastasis, and how patients get benefit from this study? Moreover, in order to enhance translational relevance, have the authors analyzed mRNA or microRNA expression profiles in tissue samples derived from patients? Additionally, have any studies been performed using primary human liver LSECs isolated from either normal or tumor-adjacent liver tissue to confirm the results seen in the cell lines?

Response 3: Thank you for these important points. We address them below:

Regarding the differential effects observed in murine versus human LSECs: while the decrease in proliferation was more pronounced in murine LSECs, the effect in human LSECs was still statistically significant. The greater response in murine cells may be attributed to differences in metabolic activity or species-specific sensitivity, but the conserved response in both models supports the relevance of PCSK9 as a target.

We believe the murine model remains valuable, as it reveals a conserved mechanism between species and provides a foundation for future in vivo studies and preclinical testing.

Although we have not yet performed full mRNA or microRNA profiling in patient-derived samples, we have included preliminary data showing results from primary human LSECs. These cells were isolated from liver tissue and show responses consistent with our in vitro cell line data. We have added this dataset to the supplementary materials Figure 1.

Reviewer 2 Report

Comments and Suggestions for Authors

The paper establishes a connection between colorectal cancer stem cells (CSCs), the upregulation of PCSK9 in liver sinusoidal endothelial cells (LSECs), and forming a pro-metastatic niche. Utilizing CSC-conditioned media to model interactions between tumors and endothelial cells demonstrates a strong methodological approach. The findings show that PF-06446846 reverses the activation of LSECs, specifically their proliferation and migration, offering a viable pathway for preventing metastasis. Consistent results from both human and murine LSECs enhance the clinical significance of targeting PCSK9. Integrating various techniques, including RNA sequencing, qPCR, Western blotting, and functional assays assessing proliferation and migration, provides a comprehensive body of evidence. This research greatly enhances our understanding of the metastatic niche and positions PCSK9 as a promising target for therapeutic intervention. Emphasizing these aspects could increase its appeal to basic science and clinical audiences.

Other comments:

The abstract references microRNAs associated with migration and proliferation; however, the results do not include specific miRNA data or pathway analysis. Although PCSK9's role in cholesterol metabolism is mentioned, the study fails to investigate whether lipid modulation influences LSEC activation.

There is no in vivo validation, such as reduction of metastasis in animal models where PCSK9 inhibition is applied. The specificity of PF-06446846 for PCSK9 compared to other proprotein convertases remains unaddressed. Additionally, there is a lack of correlation between PCSK9 levels in human metastases and patient outcomes, such as survival and treatment response.

No comparisons are made to existing anti-PCSK9 therapies, including monoclonal antibodies like alirocumab. Specific miRNAs and gene networks altered by PCSK9 inhibition need to be identified via RNA sequencing. Further research should examine PCSK9’s interaction with established LSEC receptors, such as ICAM-1 and VCAM-1.

In vivo experiments testing PF-06446846 in murine metastasis models should be included. The efficacy of PF-06446846 should also be compared to clinical-stage PCSK9 inhibitors. Sample sizes, replicates, and statistical tests used in figure legends require clarification.

  1. PF-06446846’s specificity for PCSK9 versus other proprotein convertases is not addressed.
  2. Lack of correlation between PCSK9 levels in human metastases and patient outcomes.
  3. There is no comparison to existing anti-PCSK9 therapies (e.g., monoclonal antibodies like alirocumab).
  4. Investigate PCSK9’s interaction with known LSEC receptors (e.g., ICAM-1, VCAM-1)1.
  5. Include in vivo experiments testing PF-06446846 in murine metastasis models.
  6. Compare PF-06446846’s efficacy to clinical-stage PCSK9 inhibitors.
  7. Clarify sample sizes, replicates, and statistical tests used in figure legends.
  8. Provide p-values/effect sizes for key comparisons (e.g., migration rates with/without inhibitor).
  9. Abstract: Correct "invetigate" to "investigate" and specify the CSC reprogramming method1.
  10. Figures: Include representative immunofluorescence images showing PCSK9+ LSECs in metastases.
  11. Discussion: Address how LSEC-specific PCSK9 inhibition might synergize with anti-angiogenic therapies.

Author Response

Comment 1: The abstract references microRNAs associated with migration and proliferation; however, the results do not include specific miRNA data or pathway analysis. Although PCSK9's role in cholesterol metabolism is mentioned, the study fails to investigate whether lipid modulation influences LSEC activation.

Response 1: We appreciate the reviewer’s comment and fully agree that our statements regarding microRNAs and cholesterol metabolism were not sufficiently supported by the presented data. To address this, we have revised the abstract to remove assertions linking specific microRNAs directly to proliferation and migration, as well as the reference to PCSK9’s role in cholesterol metabolism. These changes were made to maintain focus on the observed implication of PCSK9 in cancer-related processes and to avoid overinterpretation of the results. Changes were done in the abstract, page 1

Comment 2: There is no in vivo validation, such as reduction of metastasis in animal models where PCSK9 inhibition is applied. The specificity of PF-06446846 for PCSK9 compared to other proprotein convertases remains unaddressed. Additionally, there is a lack of correlation between PCSK9 levels in human metastases and patient outcomes, such as survival and treatment response.

Response 2: In vivo validation in this model is challenging, as we are specifically studying the role of PCSK9 in liver sinusoidal endothelial cells (LSECs). This would require either a conditional PCSK9 knockout mouse targeting LSECs, which is currently unavailable, or a delivery system capable of specifically targeting LSECs in vivo. Reviewer 2 mentioned PF-06446846, and we used PF-06815345. Those two molecules act in a similar way inhibiting PCSK9 at the ribosomes. Regarding the PCSK9 inhibitor, it is important to note that PCSK9 is a proprotein convertase distinct from the Furin-like PC family, and it belongs to a separate subclass within the proprotein convertase (PC) family.  PF-06446846 specificity has been documented by Nathanael G. et al. 2017, who reported that PF-06446846 does not affect other members of the PC family and PF-06815345 specificity was assessed by McClure et al. 2017 (32). Although we do not yet have data from metastasis models, Supplementary Figure 2 shows that PCSK9 expression is associated with survival outcomes in liver hepatocellular carcinoma. Additionally, studies by Wang et al. [16] have linked PCSK9 to colorectal cancer metastasis in murine models, while Weiss et al. [17] demonstrated the involvement of circulating PCSK9 in human liver metastasis.

Comment 3: No comparisons are made to existing anti-PCSK9 therapies, including monoclonal antibodies like alirocumab. Specific miRNAs and gene networks altered by PCSK9 inhibition need to be identified via RNA sequencing. Further research should examine PCSK9’s interaction with established LSEC receptors, such as ICAM-1 and VCAM-1.

Response 3: We thank the reviewer for these valuable comments. Regarding anti-PCSK9 therapies, we did consider the use of monoclonal antibodies such as alirocumab. However, the aim of our study was to specifically inhibit endogenous PCSK9 expression within liver sinusoidal endothelial cells (LSECs). Monoclonal antibodies like alirocumab primarily reduce circulating PCSK9 levels and are not effective in targeting intracellular or locally produced PCSK9 in specific cell types such as LSECs. For this reason, we chose an specific chemical inhibitor.

In response to the second point, transcriptomic data following PCSK9 inhibition are presented in Figures 2 and 4, where we show changes in gene expression associated with endothelial activation. While these findings provide initial insights, we agree that further RNA sequencing analysis—focusing on microRNAs and downstream gene networks—will be important to fully elucidate the molecular pathways involved.

Lastly, we appreciate the reviewer’s suggestion to explore PCSK9’s potential interaction with key LSEC receptors such as ICAM-1 and VCAM-1. This is an interesting direction that we will consider for future studies.

Comment 4: In vivo experiments testing PF-06446846 in murine metastasis models should be included. The efficacy of PF-06446846 should also be compared to clinical-stage PCSK9 inhibitors. Sample sizes, replicates, and statistical tests used in figure legends require clarification.

Response 4: We thank the reviewer for this important observation. In vivo validation using PF-06446846 or PF-06815345 in metastasis models presents technical limitations in our specific context. Since our study focuses on the role of PCSK9 in liver sinusoidal endothelial cells (LSECs), effective validation would require either a conditional knockout of PCSK9 in LSECs—currently unavailable—or a delivery system capable of targeting LSECs specifically in vivo. These constraints limit our ability to directly assess the impact of PF-06446846 or PF-06815345 on metastatic progression via LSEC modulation. Regarding the comparison to clinical-stage PCSK9 inhibitors: monoclonal antibodies such as alirocumab primarily target circulating PCSK9 and are not suitable for inhibiting intracellular or cell-specific PCSK9, as discussed in our previous response.

Finally, we have addressed the reviewer’s request for greater clarity on sample sizes, number of replicates, and statistical methods. These details have now been added or clarified in the figure legends on pages 6, 10, 14, and 14 of the revised manuscript.

Summary of comments (1-11):

  1. PF-06446846’s specificity for PCSK9 versus other proprotein convertases is not addressed. Responded comment 2.
  2. Lack of correlation between PCSK9 levels in human metastases and patient outcomes. Responded in comment 2.
  3. There is no comparison to existing anti-PCSK9 therapies (e.g., monoclonal antibodies like alirocumab). Responded in comment 3.
  4. Investigate PCSK9’s interaction with known LSEC receptors (e.g., ICAM-1, VCAM-1)1. Responded in comment 3.
  5. Include in vivo experiments testing PF-06446846 in murine metastasis models. Responded in comment 2 and 4.
  6. Compare PF-06446846’s efficacy to clinical-stage PCSK9 inhibitors. Responded in comment 4.
  7. Clarify sample sizes, replicates, and statistical tests used in figure legends. Responded in comment 4.
  8. Provide p-values/effect sizes for key comparisons (e.g., migration rates with/without inhibitor). We have provided the new changes in pages 6, 10, 14, and 14
  9. Abstract: Correct "invetigate" to "investigate" and specify the CSC reprogramming method1. It has been corrected in the abstract page1.
  10. Figures: Include representative immunofluorescence images showing PCSK9+ LSECs in metastases. It has been added in Figure 3-E.
  11. Discussion: Address how LSEC-specific PCSK9 inhibition might synergize with anti-angiogenic therapies. It has been included in the discusión page 17.

Reviewer 3 Report

Comments and Suggestions for Authors

The following corrections/clarifications are required. The comments are being provided section-wise:

One of the main limitations of the study, from the point of view of this reviewer, which should be pointed out in the introduction, is the aim is not expressed in an intelligible way according to hypothesis.

The content of the manuscript, Front matter, Research manuscript sections an Back matter sections are not prepared in accordance with the journal format. The "Instructions for Authors" section of the journal should be reviewed and edited.

The manuscript should be prepared according to the journal template.

The quality of images (figures) should be improved.

Limitations of the current study should be added to the discussion section. The conclusion section is incomplete and this section should be added to highlight important parts.

References should be rearranged and written in the appropriate format.

References should be updated. Especially the discussion section should be compared with current referencess and the results should be discussed.

Author Response

Comments 1: One of the main limitations of the study, from the point of view of this reviewer, which should be pointed out in the introduction, is the aim is not expressed in an intelligible way according to hypothesis.

Response 1: Thank you for pointing out this issue; the aim has been clarify according to the hypothesis in page 2.

Comment 2: The content of the manuscript, Front matter, Research manuscript sections an Back matter sections are not prepared in accordance with the journal format. The "Instructions for Authors" section of the journal should be reviewed and edited. The manuscript should be prepared according to the journal template.

Response 2. This point has been corrected.

Comment 3: The quality of images (figures) should be improved.

Response 3: They have been re-upload at 600 dpi to improve the quality and figure 2B was modified for a better understanding of the genes names.

Comment 4: Limitations of the current study should be added to the discussion section. The conclusion section is incomplete and this section should be added to highlight important parts.

Response 4: It has been added in discussion page 16

Comment 5: References should be rearranged and written in the appropriate format. References should be updated. Especially the discussion section should be compared with current references and the results should be discussed.

Response 5: More recent references were added in the discussion page 16-17

Round 2

Reviewer 2 Report

Comments and Suggestions for Authors

The authors addressed all the comments provided by the reviewers, enhancing the quality of the manuscript. It is now ready for consideration for publication.

Reviewer 3 Report

Comments and Suggestions for Authors

I think that the authors have adequately addressed the comments made by the reviewers in the revised version of the manuscript. Therefore, I have no further comments.